Review

Subject Area:
cellular biology/biochemistry

Keywords:
endoplasmic reticulum-associated degradation, MARCH E3 ligases, protein degradation, serine/threonine ubiquitination, ubiquitin–proteasome system

Authors for correspondence:
Amie J. McClellan
e-mail: amcclellan@bennington.edu
Lars Ellgaard
e-mail: lellgaard@bio.ku.dk

# Cellular functions and molecular mechanisms of non-lysine ubiquitination

Amie J. McClellan[1], Sophie Heiden Laugesen[2] and Lars Ellgaard[2]

[1]Division of Science and Mathematics, Bennington College, 1 College Drive, Bennington, VT 05201, USA
[2]Department of Biology, Linderstrøm-Lang Centre for Protein Science, University of Copenhagen, 2200 Copenhagen N, Denmark

 AJM, 0000-0002-8033-6748; LE, 0000-0002-7018-0137

Protein ubiquitination is of great cellular importance through its central role in processes such as degradation, DNA repair, endocytosis and inflammation. Canonical ubiquitination takes place on lysine residues, but in the past 15 years non-lysine ubiquitination on serine, threonine and cysteine has been firmly established. With the emerging importance of non-lysine ubiquitination, it is crucial to identify the responsible molecular machinery and understand the mechanistic basis for non-lysine ubiquitination. Here, we first provide an overview of the literature that has documented non-lysine ubiquitination. Informed by these examples, we then discuss the molecular mechanisms and cellular implications of non-lysine ubiquitination, and conclude by outlining open questions and future perspectives in the field.

## 1. Introduction

### 1.1. The ubiquitin–proteasome system

Ubiquitin is a 76-amino-acid polypeptide that is highly conserved among eukaryotes. The covalent attachment of ubiquitin, namely *ubiquitination*, plays a crucial role in many biological pathways, such as DNA damage repair, signal transduction, inflammatory response generation, endocytosis, transcriptional regulation and cell-cycle progression [1]. Perhaps the best-studied purpose of ubiquitin attachment, however, is to signal for proteasome-mediated protein degradation. In this so-called ubiquitin–proteasome system (UPS), attachment of ubiquitin moieties targets substrate proteins for degradation by the 26S proteasome, a 2.5-MDa molecular machine that first recognizes substrates through their ubiquitin modification; this is followed by substrate deubiquitination, unfolding and ultimately degradation in a central proteolytic chamber.

The 26S proteasome consists of the 20S proteolytic core and two attached 19S regulatory complexes. The regulatory complexes recognize ubiquitinated substrates using ubiquitin receptors, i.e. ubiquitin-interacting proteins. The 19S cap also contains two (*Saccharomyces cerevisiae*) or three (most other eukaryotes) deubiquitinating enzymes (DUBs); over 100 DUB-encoding genes have been predicted in the human genome to date [2,3]. Ubiquitin is synthesized as an inactive precursor of multiple ubiquitin units, and DUBs play an important function in cleaving the precursor to release free ubiquitin [4]. In the context of the proteasome, DUB-catalysed removal of substrate ubiquitin units permits ubiquitin to be recycled for further use. Moreover, ubiquitin removal by DUBs may provide a last-minute opportunity for a protein to escape degradation [5,6]. Following ubiquitin removal, the substrate protein is unfolded by a ring of AAA (ATPases Associated with diverse cellular Activities) ATPases located at the entry of the proteolytic core. Finally, the unfolded polypeptide chain is threaded through a narrow channel into the central cavity of the 26S proteasome, where proteases ensure its efficient cleavage into peptides [6,7].

## 1.2. The ubiquitination process

Ubiquitination occurs through the sequential action of three enzyme classes: ubiquitin-activating (E1), ubiquitin-conjugating (E2) and ubiquitin ligase (E3). Upon processing of the ubiquitin precursor by DUBs that cleave specifically after the C-terminal Gly75-Gly76 motif, Gly76 is exposed and poised to react. Ubiquitin becomes 'activated' when an E1 enzyme uses the energy supplied by ATP hydrolysis to generate a thioester bond between its active-site cysteine residue and the C-terminus of ubiquitin. In the next step, ubiquitin is transferred onto the active-site cysteine of an E2 enzyme (figure 1*a*). The charged E2 finally interacts with an E3 ligase that facilitates positioning and transfer of ubiquitin onto the substrate by one of several mechanisms dependent on the class of E3 enzyme (see below) [8]. Irrespective of the class of E3 enzyme, however, it is the E3 that determines substrate recognition and thus specificity in the system.

## 1.3. E2 enzymes and E3 ligases: a brief overview

E2 enzymes (of which approx. 40 have been found in the human proteome) are characterized by a conserved core region of approximately 150 amino acid residues. This region accounts for the enzymatic activity of the E2s due to a catalytic cysteine residue and is referred to as the ubiquitin-conjugating (UBC) domain. In the cell, E2s primarily exist as E2-ubiquitin conjugates (figure 1*a*), prone to ubiquitinate their substrates [9–11]. However, E2-ubiquitin conjugates show low rates of ubiquitin transfer in the absence of an E3 enzyme [10]. Binding studies of E2-E3 pairs have recently shown that E2s have a markedly higher affinity for their cognate E3 when the E2 is conjugated to ubiquitin [12].

E3 enzymes are by far the most numerous enzyme family in the ubiquitination cascade, represented by approximately 700 enzymes in mammals. E3s are classified into subfamilies according to the presence of either a RING (Really Interesting New Gene) finger or a HECT (Homologous to the E6AP Carboxyl Terminus) domain. E3 ligases of the RING finger family can simultaneously bind ubiquitin-charged E2 and substrate, and thus facilitate the direct transfer of the ubiquitin moiety onto the substrate. On the other hand, the non-RING E3 ligases of the HECT type, as well as the RING-between-RING (RBR)-type ligases, constitute a molecular stepping stone for the ubiquitin molecule through a covalent interaction with an active-site cysteine of the E3 enzyme [13,14]. By this mechanism, ubiquitin is transferred from the E2 to the E3 and from here onto specific substrate acceptor sites [15,16].

## 1.4. Ubiquitin linkage types can determine substrate fate

Sequential rounds of ubiquitination yield polyubiquitinated protein substrates. During ubiquitin chain elongation, the carboxyl group of the C-terminal Gly76 within the preceding ubiquitin molecule is linked via an isopeptide bond to one of the seven lysine residues (Lys6, Lys11, Lys27, Lys29, Lys33, Lys48 or Lys63) of the subsequent ubiquitin moiety. Furthermore, head-to-tail ubiquitin conjugation occurs, where the C-terminal Gly76 of the proximal ubiquitin molecule is linked to the N-terminal amino group of Met1 of the distal ubiquitin via a regular peptide bond [1,17]. Ubiquitin chains can be homotypic (involving the same lysine residue), heterotypic (involving different lysine residues in one chain) or branched, where a ubiquitin molecule is further ubiquitinated on different lysine residues.

It has been generally accepted that Lys48-linked polyubiquitin chains of four ubiquitin moieties or more are required for proteasomal degradation [18]. However, other chain topologies can mediate degradation. Studies of *S. cerevisiae* have shown that Lys11 linkages preferentially modify a subset of protein substrates and play an important role in endoplasmic reticulum-associated degradation (ERAD; §2.1). Additionally, it has been shown that branched Lys11/Lys48-linked chains provide a better signal for degradation by the proteasome than do non-branched chains [19]. Even monoubiquitination on one or more sites can be sufficient for proteasomal recognition and degradation [20,21].

## 1.5. Non-lysine ubiquitination

For many years, lysine was considered the only site for substrate ubiquitination, and it is still considered the canonical ubiquitination site. However, it is now established that cysteine, serine and threonine residues, as well as the free amino group of the N-terminus of proteins, also function as sites for ubiquitination, forming thioester, hydroxyester and peptide bonds, respectively (figure 1*a*) [22]. Although the tyrosine side chain is also hydroxylated, like serine and threonine, no examples of ubiquitination occurring on tyrosine exist to date. This may not be surprising, given the presence of the benzene ring, leaving the oxygen lone pairs less reactive that those on serine or threonine. Finally, some bacteria have evolved a ubiquitination mechanism, carried out by proteins of the SidE effector family, that results in phospho-ribosyl-linked ubiquitin conjugated to serine residues of the protein substrate (figure 1*b*) [23]. In this review, we focus on non-lysine ubiquitination as it pertains to the conjugation of ubiquitin to the side chains of cysteine, serine or threonine, and not on the N-terminal ubiquitination of a substrate or ubiquitination performed by the SidE effectors.

The conjugates generated from ubiquitination on non-lysine residues are thermodynamically less stable than the ones generated on lysine residues. This raises interesting questions concerning their different physiological functions (§§4.1 and 4.2), which may be governed in part by differences in reaction kinetics.

The thioester bond generated from conjugating ubiquitin onto a cysteine residue is the least stable bond that forms between ubiquitin and a substrate protein. As described above, ubiquitin is transferred from an E1 onto an E2 enzyme and finally (via an E3 enzyme in the case of non-RING-type E3s) onto the substrate. Thus, intermediate ubiquitin-E1/E2/E3 conjugates are linked via a thioester bond formed between Gly76 of ubiquitin and a cysteine residue of the enzyme. Because the sulfur lone pair is a soft nucleophile that reacts preferentially with soft electrophiles, the reaction with a thioester carbonyl (i.e. the electrophile of the thioester linkage on the E2) is faster than the reaction of a hydroxyl, which is, in turn, faster than the reaction of an amine group with the E2 thioester carbonyl [24]. Therefore, even if ubiquitination on a cysteine residue results in a relatively weak bond, this thioester bond is readily formed. In signalling events where a rapid response is required, this

**Figure 1.** Schematic overview of canonical and non-canonical ubiquitination. (*a*) The lone pair of a substrate nucleophile, X, representing lysine/N-terminal amines, cysteine thiols or serine/threonine hydroxyls, attacks the electrophile thioester carbonyl of an E2–ubiquitin (Ub) conjugate. This results in a Ub-substrate conjugate linked by an isopeptide (lysine), peptide (N-terminal), thioester (cysteine) or hydroxyester bond (serine/threonine), respectively. Here, we focus on non-lysine ubiquitination defined as conjugation of Ub on the side chains of cysteine, serine or threonine (as indicated by the box). (*b*) Ubiquitin conjugated to a substrate as a result of ubiquitination mediated by SidE effector proteins. A phosphoribosyl links Arg42 of Ub to a substrate serine residue.

type of ubiquitination may be preferable. Physiologically, the combination of lysine and non-lysine ubiquitination endows the UPS with greater flexibility.

# 2. Non-lysine ubiquitination targets a variety of cellular processes

Previous reviews have provided thoughtful insight into non-lysine ubiquitination based on the literature known at the time [24,25]. First identified as an activity of virus-encoded E3 ligases affecting the stability of components of the human immune system [26–28], the process has since been demonstrated in yeast and at various cellular locations. Moreover, examples of non-lysine ubiquitination actors and substrates are rapidly increasing. Below, we review the existing literature on non-lysine ubiquitination (with the focus defined above), organized according to the cellular processes involved. Table 1 summarizes known substrates, as well as E2 and E3 enzymes implicated in non-lysine ubiquitination.

## 2.1. Endoplasmic reticulum-associated degradation (ERAD)

Much of the crucial information about non-lysine ubiquitination has been obtained from studies of ERAD [55]. When folding or oligomeric assembly of proteins in the ER goes awry, misfolded proteins accumulate [56]. Aberrant proteins are first recognized by ER chaperones and/or specific ERAD factors. They are then targeted to the cytosol in a

process termed dislocation that involves a channel through which the substrate is retrotranslocated. Once an ERAD substrate reaches the cytosolic side of the ER membrane, membrane-embedded E3 ligases work together with E2 enzymes to effect polyubiquitination. Finally, the 26S proteasome degrades the protein. Several viruses usurp the ERAD system to avoid immune detection, often by targeting important molecules of the immune system, such as the major histocompatibility complex (MHC) class I heavy chain (HC), for degradation. While this degradation is initiated at the ER membrane in ERAD, MHC-I HC can also undergo ubiquitination-dependent endocytosis at the plasma membrane followed by lysosomal degradation (see §2.3). Not only viruses exploit the ERAD pathway for their own purposes—a range of AB-type toxins, such as cholera, ricin and pertussis, use the ERAD pathway to gain access to the cytosol, where they perform their deleterious functions ([57]; §4.1).

### 2.1.1. MHC class I heavy chain and CD4

The first known example of an ERAD substrate modified by non-lysine ubiquitination was MHC-I HC. The mouse γ-herpesvirus protein, mK3, is an E3 ligase that directs MHC-I HC for degradation in infected cells. The C-terminal cytosolic tail of MHC-I HC, a type I transmembrane protein, contains both lysine and serine residues. Early work showed that although this cytosolic tail is required for degradation of the protein, degradation of a mutant of MHC-I HC with all tail lysines replaced by arginines proceeded efficiently in cells expressing mK3 [58]. While these findings may be explained by a model

**Table 1.** Overview of the literature documenting non-lysine ubiquitination, with a focus on internal cysteine, serine and/or threonine residues. ND, not determined. Notations in the 'modified residues' column convey the following: 'Non-Lys' indicates that the ubiquitination could be N-terminal, or on an internal non-lysine residue (cysteine, serine and threonine); e.g. a lysine-free version of the substrate was still degraded, but no further experimental evidence regarding what residue(s) is/are ubiquitinated was provided. 'Not N-terminal' indicates that N-terminal ubiquitination was ruled out experimentally. 'Lys/Cys/Ser/Thr' (in various combinations) indicates that there is indirect (i.e. mutation of other residue(s) does not prevent ubiquitination and/or degradation), semi-direct (i.e. ubiquitination sensitive to reducing agents or alkaline treatment) or direct (mass spectrometry) evidence that ubiquitination can occur on those residues. If known, preference for certain residues over others is indicated. The ± designation indicates that a substrate was tested both in its native state (with endogenous lysines or N-terminus available) and in a recombinant form (with endogenous lysines mutated to arginines or the N-terminus chemically blocked, for example).

| E2(s) | E3(s) | substrate(s) | modified residue(s) | process | reference(s) |
|---|---|---|---|---|---|
| ND | Mir1/kK3 | Lys-free MHC-I HLA.B7 Lys/Cys-free MHC-I HLA.B7 | non-Lys, Cys non-Lys, non-Cys | virus-induced endocytosis/ degradation | [26] |
| ND | mK3 | MHC-I HC ± Lys MHC-I HC ± N-Ub | Ser/Thr | virus-induced ERAD | [27] |
| Ube2J2 | mK3 | MHC-I HC variants | preferential Ub of Ser/Thr | virus-induced ERAD | [28] |
| ND | SCFβ-TrCP (cellular) with Vpu (viral) | CD4 | Lys/Ser/Thr (Cys not investigated) | virus-induced ERAD | [29,30] |
| ND | ND | Lys-free TCRα | non-Lys | ERAD | [31] |
| Ube2G2 | ND | Lys-free TCRα | non-Lys | ERAD | [32] |
| ND | HRD1 | TCRα | Ser267, Ser268 (Cys, Lys, Thr are able to substitute) | ERAD | [33] |
| ND | HRD1 (for all variants) | NS-1 κ LC variants | Ser/Thr preferred over Lys; not N-terminal; non-Cys | ERAD | [34] |
| ND | HRD1 | mini-HC and NHK α1-anti-trypsin | Ser/Thr | ERAD | [34] |
| ND | HRD1 (± Lys) | TCRα ± Lys | non-Lys; non-Cys; not N-terminal | ERAD | [35] |
| ND | ND | neutrophil elastase (naturally Lys-free) | non-Lys | ERAD | [35] |
| Ube2J1 | HRD1 (±Lys) | MHC-I HC ± Lys | Lys preferred over Ser/Thr | ERAD | [36] |
| Ubc6/Ubc7 (Ube2J2/ Ube2G2 homologues) | DOA10 (MARCH6 homologue) | Vma12 Sbh2 Ubc6 | Ubc6 mono-Ub Lys/Ser*/Thr | ERAD | [37] *Ub shown by mass spec |
| ND | ND | SM/SQLE (N100 degron) | non-Lys | regulated ERAD | [38] |
| Ube2j2 | MARCH6 | SM/SQLE (N100 degron) | Ser59, Ser61, Ser83*, Ser87 | regulated ERAD | [39] *Ub shown by mass spec |
| ND | ND | Bid-N (Lys-free amino terminus) | Cys, Ser/Thr; not N-terminal | regulated degradation | [40] |
| ND | ND | N-terminally blocked, Lys-free neurogenin2 | Cys, Ser/Thr | regulated degradation | [41] |

(*Continued.*)

royalsocietypublishing.org/journal/rsob  Open Biol. 9: 190147

**Table 1.** (Continued.)

| E2(s) | E3(s) | substrate(s) | modified residue(s) | process | reference(s) |
|---|---|---|---|---|---|
| ND | ND | N-terminally blocked, Lys-free neurogenin3 | Cys, Ser/Thr | regulated degradation | [42] |
| dBRUCE | ND | reaper ± Lys | non-Lys | regulated degradation | [43] |
| Ubc6/Ubc7 (Ube2J2/Ube2G2 homologues) | DOA10 (MARCH6 homologue) | Asi2 ± Lys | Ser/Thr in Lys-free, Lys preferred in WT | regulated degradation | [44] |
| ND | ND | NY-ESO-1 antigen ± Lys | Ser/Thr, not N-terminal | antigen processing/presentation | [45] |
| Ube2D1 | ND | Lys-free MARCH1 | non-Lys | regulated endocytosis | [46] |
| Ube2D1 Ube2D3 Ube2E1 | MYCBP2/Phr1 | Free amino acids; pentapeptides | E3 prefers Thr over Ser | in vitro | [47] |
| ND | MYCBP2/Phr1 | NMNAT2 | Ser/Thr, non-Cys | in vitro | [47] |
| Pex4 (Ubc10) | ND | S. cerevisiae Pex5 (N-terminally blocked, Lys-free) | Cys6 | peroxisomal signal receptor recycling | [48] |
| ND | ND | mammalian Pex5 | Cys11 | peroxisomal signal receptor recycling | [49] |
| UbcH5a UbcH5b UbcH5c | ND | mammalian Pex5 | Cys11 | peroxisomal signal receptor recycling | [50] |
| Pex4 (Ubc10) | Pex12 | S. cerevisiae Pex5 | Cys6 | peroxisomal signal receptor recycling | [51] |
| ND | ND | S. cerevisiae Pex18 | Cys6 | peroxisomal signal receptor recycling | [52] |
| Pex4 (Ubc10) | Pex10, Pex12 | S. cerevisiae Pex18 | Cys6 | peroxisomal signal receptor recycling | [53] |
| Pex4 | Pex2, Pex10, Pex12 | P. pastoris Pex20 (Pex18 homologue) | Cys8 | peroxisomal signal receptor recycling | [54] |

whereby partial dislocation of the ectodomain would result in its ubiquitination upon exposure to the cytosol, subsequent work demonstrated that MHC-I HC entirely devoid of lysines was still ubiquitinated and efficiently degraded [27]. Moreover, it was shown that mK3 could facilitate ubiquitination of a single serine, threonine or lysine (but not cysteine) residue placed close to the C-terminus of the cytosolic tail of MHC-I HC, resulting in its efficient degradation [27]. These surprising findings were extended to show that, even in the context of the native sequence, serine residues were preferentially ubiquitinated over lysine residues in a reaction catalysed by the E2 Ube2j2, a tail-anchored ER membrane protein, which was shown to be the cognate E2 of mK3 [28]. Thus, despite the presence of several lysine residues in the MHC-I HC tail region, ubiquitin was primarily conjugated to serine/threonine residues, as judged by the notable loss of ubiquitinated species upon NaOH treatment.

This experimental procedure is used to probe for serine/threonine ubiquitination, due to the alkaline sensitivity of the relatively labile hydroxyester bond. In similar work, both lysine and serine/threonine residues in the cytosolic tail of the immune receptor CD4 can be ubiquitinated and contribute to the ERAD of CD4 in cells expressing Vpu, an accessory type I integral membrane protein expressed by HIV-1 [29]. To accomplish the rapid degradation of CD4 molecules from the ER membrane, Vpu recruits a cellular E3 ligase complex, SCFβ-TrCP [29].

The human cytomegalovirus encodes two proteins, US2 and US11, that induce rapid cellular degradation of MHC-I HC molecules although neither is an E3 ligase [59]. Like mK3-mediated degradation, US11-mediated degradation of lysine-free MHC-I HC proceeds efficiently, indicating non-lysine ubiquitination [60]. While non-lysine ubiquitination of the MHC-I HC cytosolic tail was not directly

demonstrated, recent work identified that a previously uncharacterized E3 ligase, TMEM129, uses Ube2j2 as its cognate E2 in the US11-mediated degradation of MHC-I HC [61,62]. In addition, another E2 enzyme, Ube2k, was shown to be important for ubiquitination [62]. Taken together, these data are compatible with a model where TMEM129 recruits Ube2j2 to facilitate ubiquitination on serine and/or threonine residues on the cytosolic tail of MHC-I HC. Moreover, it was suggested that Ube2j2 functions to initiate substrate ubiquitination, whereas Ube2k, which itself is not able to initiate ubiquitination, extends the monoubiquitinated substrate with Lys48-linked ubiquitins [62,63].

### 2.1.2. T-cell receptor complex

Another protein of the immune system, the α-subunit of the T-cell receptor complex (TCRα), is an important model substrate in ERAD studies. This stems from the finding that, when present in stoichiometric excess of its companion subunits, TCRα is rapidly degraded [64]. In analogy to the studies on MHC-I HC, early work showed that lysine-free TCRα is efficiently degraded in a proteasome-dependent manner [31]. In a stable isotope labelling in cell culture approach, lysine-free TCRα was shown to be ubiquitinated on serine/threonine residues as judged by sensitivity to alkaline treatment, although mass spectrometry (MS) did not succeed in identifying the modified residues [35]. By contrast, three specific ubiquitinated lysine residues were identified in the wild-type protein, which did not show evidence of non-lysine ubiquitination.

This result seemingly contrasts the finding that two conserved serine residues present at the very C-terminal end of the short cytosolic tail of TCRα mediate ubiquitination and ERAD of the protein [33]. In mutational studies, it was found that substitution of the two serines with alanine residues almost completely prevented ubiquitination and stabilized TCRα. Conversely, insertion of additional serine residues in the tail region increased ubiquitination as well as the kinetics of degradation. Finally, replacement of the two serine residues with threonine, cysteine or lysine residues still allowed ubiquitination and fast degradation. While the formal possibility exists that the serine-mediated TCRα degradation may be an indirect effect of the serine residues promoting ubiquitination of other residues in the protein, the most likely explanation is direct ubiquitination of these residues. Concerning the involved ubiquitination machinery, the E3 ligases Hrd1 [33,35] and gp78 [65] have been shown to mediate TCRα ubiquitination, whereas E2 enzymes Ube2j1 [66], Ube2j2 [66] and Ube2g2 [32,65] have all been implicated in the process. Notably, unassembled TCRα chains have been demonstrated not to integrate efficiently into the membrane, but instead localize to the ER lumen [67]. In essence, these chains can, therefore, be considered as soluble ERAD substrates (see below). It is interesting to note that the two very C-terminal (serine) residues become ubiquitinated, indicating that this substrate is likely to be retrotranslocated with the C-terminus first and ubiquitinated as soon as it enters the cytosol.

### 2.1.3. Soluble ERAD substrates

All examples above involve transmembrane proteins with non-lysine ubiquitination occurring on the cytosolic portion of the targeted molecule. However, there is evidence that the soluble ERAD substrate non-secreted immunoglobulin κ light chain (NS-1 κ LC) is also extensively ubiquitinated on serine/threonine residues [34]. Using NaOH treatment, it was established that NS-1 κ LC is mainly ubiquitinated on serine and threonine residues. However, lysine residues are also modified since polyubiquitination was only reduced substantially when mutating all serines, threonines and lysines, correlating with significant cellular stabilization of only this mutant. Moreover, two other soluble ERAD substrates were also found to contain ubiquitination that was labile under alkaline conditions [34]. As observed for both TCRα and MHC-I HC [33,35,36], Hrd1 activity was critical for ubiquitination and degradation of the three soluble ERAD substrates tested.

## 2.2. Regulated degradation

Regulated protein degradation by the UPS is essential to precisely control levels of proteins during development and differentiation, as well as in central cellular processes such as cell-cycle control or regulation of metabolism and biosynthesis. Below are examples of non-canonical ubiquitination in pathways that rely on a finely tuned balance between protein components.

### 2.2.1. Cholesterol biosynthesis

Cholesterol homeostasis and biosynthesis are governed in part by two rate–limiting enzymes: 3-Hydroxy-3-Methylglutaryl-CoA Reductase (HMGCR) and squalene monooxygenase (known as SQLE or SM) [68]. SQLE stability is inversely correlated with cellular cholesterol levels, as SQLE is subjected to regulated ERAD when additional cholesterol synthesis is unnecessary.

Several examples in the literature connect SQLE degradation to E2 and E3 enzymes implicated in non-lysine ubiquitination. In yeast, the SQLE homologue Erg1 exhibits regulated degradation in response to plentiful sterols; this requires the E2 Ubc6 and the Doa10 E3 ligase, which also participate in non-regulatory ERAD (see §3.1.1). The replacement of a cluster of four lysine residues with arginines resulted in near-complete stabilization of Erg1, suggesting that alternative residues are unused or unavailable in this case [69]. The mammalian homologues of Ubc6 and Doa10, namely Ube2j2 and membrane-associated RING finger protein 6 (MARCH6), have been linked to SQLE degradation [70–72], but there are some clear differences. First, Erg1 residue Lys311, determined to be the critical lysine for its degradation, is not conserved in SQLE. In addition, Erg1 lacks an N-terminal sequence known as SM N100, which serves SQLE both as a cholesterol sensor and instability-conferring degron [38]. Of note, the transferable SM N100 degron still behaves as such even when all five of its lysine residues are mutated to arginine [38]. Recently, a careful investigation of SM N100 serine residues identified Ser59, Ser61, Ser83 and Ser87 as critical for cholesterol-induced degradation and provided direct MS-based evidence that Ser83 is a site for ubiquitination; the evidence in this study also suggests that both MARCH6 and catalytically active Ube2j2 are important for this process [39]. Overall, these studies support that SQLE is capable of undergoing non-lysine-dependent degradation, which is facilitated by Ube2j2 and MARCH6.

### 2.2.2. Apoptosis

Regulated degradation involving non-lysine residues is also seen in proteins linked to apoptotic pathways. The *Drosophila* protein REAPER serves as a pro-apoptotic factor for regulated apoptosis during development or in response to DNA damage. The genetically associated protein dBRUCE, which encodes an E2 Ubc domain, was found to downregulate REAPER levels regardless of whether its lysines were mutated [43]; this provides a basis for the prior observation that dBRUCE overexpression led to diminished apoptosis even when REAPER lacked lysines [73].

A second example is the Bid protein, a Bcl-2 family member with a normally sequestered pro-apoptotic domain. In response to appropriate death receptor signalling pathways, Bid is cleaved into two portions; the C-terminal region (tBid-C) goes on to signal for apoptosis, made possible by the ubiquitination and degradation of the inhibitory N-terminal fragment (tBid-N), which remains associated post-cleavage. This regulatory degradation is lysine-independent and is not due to N-terminal ubiquitination of tBid-N. Instead, chemical and mutational analyses suggest that ubiquitination both on cysteine as well as on serine/threonine residues is critical [40].

### 2.2.3. Neuronal degeneration

Neuronal injury may result in the detachment of an axon from its cell body, thus necessitating its destruction; axonal destruction is also observed in various neurodegenerative diseases. This process requires, in part, low levels of Nicotinamide Mononucleotide Adenyltransferase (NMNAT2), which is targeted for proteasomal degradation by the MYCBP2 (Myc Binding Protein 2) E3 ligase. Recent *in vitro* experiments demonstrate that MYCBP2 preferentially discharges ubiquitin from affiliated E2s to threonine residues, which were favoured approximately 10-fold over serines in experiments comparing the effects of free amino acids [47] (see also §3.3). Additionally, MYCBP2-dependent ubiquitination of NMNAT2 *in vitro* was largely diminished by alkaline treatment, suggesting that this behaviour extends to actual substrates and not only to free amino acids [47].

### 2.2.4. Asi2

As described below in §3.1, the E2s Ubc6 and Ubc7 can work together with the Doa10 E3 to ensure efficient degradation of certain ERAD substrates in *S. cerevisiae* [37]. They also perform this task for lysine-free Asi2, a transmembrane protein of the inner nuclear membrane that functions as part of a complex responsible for the quality control of misfolded or mislocalized inner nuclear membrane proteins [74,75]. Additionally, the Asi complex negatively regulates transcription factors critical for responding to the presence of available extracellular amino acids. Asi2 itself is targeted for proteasomal degradation to alleviate this repression.

The wild-type Asi2 sequence contains 10 lysine residues, all exposed to the nucleus, which are efficiently ubiquitinated. Alkaline treatment did not influence the level of wild-type ubiquitination, indicating that non-lysine ubiquitination is insignificant for the endogenous protein [44]. Interestingly, the degradation of lysine-free Asi2 proceeded with similar kinetics to the wild-type protein and the mutant protein showed robust

polyubiquitination that was sensitive to NaOH treatment. In addition to a combined total of 42 serine or threonine residues, Asi2 contains a single cysteine; polyubiquitination, however, was not abrogated by reducing agent. Regarding the relevant E2 and E3 enzymes, both wild-type and lysine-free Asi2 were stabilized in *S. cerevisiae* strains deleted for Doa10, Ubc6 and Ubc7 [44]. Whether Ubc6 and Ubc7 work in tandem during the degradation of Asi2, as is the case in the degradation of Sbh2 (§3.1), is presently unknown.

## 2.3. Virus-induced degradation via endocytosis

As mentioned above, there are numerous examples of virus-induced degradation of cellular proteins. The primary targets are immune system molecules en route to, or already on duty in, the plasma membrane. In fact, the first evidence that amino acid residues other than lysine can be ubiquitinated came from studies of a viral E3 ligase and its induced degradation of MHC-I HCs from the plasma membrane [26].

The Kaposi's sarcoma-associated herpesvirus-encoded Mir1/kK3 E3 could promote the degradation of lysine-free MHC-I HC, suggested to occur via ubiquitination of a conserved cysteine. The observed ubiquitination was sensitive to reducing agent, indicative of a thioester bond. However, the experiment was conducted at pH 11, which would also affect hydroxyester bonds. Furthermore, the authors showed that the introduction of a lysine, cysteine or serine into a poly-glycine version of the heavy chain's cytosolic tail was sufficient to restore degradation [26].

Vpu, again acting with SCFβ-TrCP (see §2.1.1), also interacts with BST-2 (also known as tetherin or CD317), a protein that inhibits viral egress from cells. BST-2 levels are downregulated in response to Vpu via endocytosis and lysosomal degradation; mutational analysis of the cytosolic tail of BST-2 found that ubiquitination still occurred even if all lysines, serines and threonines were mutated, although *in vitro* assays also demonstrated that ubiquitination of a lysine-free version of the tail was sensitive to alkaline treatment, suggesting that hydroxyester linkages can occur [76]. This conclusion is supported by earlier work that showed the most effective downregulation of BST-2 depended upon a consecutive serine–threonine–serine sequence in its cytosolic domain [77].

Taken together, the available data show that the viral manipulation of cellular proteins that stave off successful infection often use non-lysine ubiquitination strategies. This includes virally encoded E3 ligases acting with cellular E2s, or virally encoded accessory proteins that recruit cellular E3 ligases to the task (as described in §2.1.1).

## 2.4. Additional examples

### 2.4.1. Neurogenins

The neurogenins are a family of bHLH transcription factors with roles not only in neurogenesis but also in endocrine development of the pancreas and gut and in spermatogenesis. While N-terminal ubiquitination is sufficient to target *Xenopus* neurogenin2 (Ngn2) for degradation in both *Xenopus* extracts and mammalian cells, experiments with modified versions of Ngn2 lacking lysines and incapable of N-terminal ubiquitination identified cysteines as sites for ubiquitination [41,78]. When cysteines were also removed, however, Ngn2 was still polyubiquitinated. Combined with the demonstration

that alkaline treatment disrupted the observed ubiquitination, this supports that Ngn2 may also be ubiquitinated on serine or threonine [41]. Importantly, all types of ubiquitin conjugates were sufficient to support the proteasomal degradation of Ngn2, although in perhaps more physiologically relevant conditions the lack of N-terminus or lysine availability drastically stabilizes the protein, suggesting that they are critically important for rapid degradation under normal circumstances [78]. Of note, while ubiquitination on non-lysine internal residues has also been observed for the related protein Ngn3, cysteine ubiquitination was unable to target the protein for degradation while, for lysine-free Ngn3, serine/threonine ubiquitination could [42]. The identities of the cellular E2 and E3 enzymes involved in Ngn ubiquitination are presently unknown.

### 2.4.2. Peroxisome import receptors

While we have previously discussed ubiquitination on cysteines mediated by virally encoded E3 ligases (§2.3), the first demonstrations that cellular proteins could ubiquitinate cysteines came from studies of peroxisomal signal receptor proteins in yeast and mammalian model systems [48,49]. The targeting of proteins for import into the peroxisomal matrix is accomplished by two different peroxisome targeting signals and pathways, PTS1 and PTS2, each of which has its own specific receptor (reviewed in [79]).

Pex5, the requisite receptor for PTS1, may be polyubiquitinated for proteasomal degradation when it has outlived its useful lifespan or is defective, but is also transiently mono-ubiquitinated for regulatory purposes at the peroxisomal membrane. The former involves the Ubc4 E2, while the latter is due to Pex4 (aka Ubc10), or, in mammals, to UbcH5a, b or c isoforms (figure 2a) [50]. In addition to using different E2s, each pathway involves ubiquitination on different residues. While Ubc4 transfers ubiquitin to lysine, Pex4 makes use of a conserved cysteine for ubiquitin conjugation. This transient ubiquitin signalling seems important for continuing Pex5 recycling and activity rather than its degradation as cysteine ubiquitination does not affect the stability of Pex5 [80]. While replacing the conserved cysteine residue with lysine still permits recycling, the resultant polyubiquitination increases the likelihood of degradation over further recycling and function (figure 2b(ii)) [50,80].

In *Pichia pastoris*, Pex20, along with Pex7, constitutes the signal receptor for the PTS2 pathway targeting proteins for import into the peroxisomal matrix. Similar to what is seen for Pex5, the functional recycling of Pex20 requires a conserved cysteine [81], which was later confirmed to be ubiquitinated using Pex4 [54]. The *S. cerevisiae* homologue of Pex20, Pex18, is also ubiquitinated on this conserved cysteine by Pex4 [52,53].

The identities of the relevant E3 ligases have also been established; peroxisome-localized RING-domain-containing proteins appear to serve as the E3s for Pex5 ubiquitination. While all three RING peroxins, Pex2, Pex10 and Pex12, form a heteromeric complex [82], they are capable of distinct functions. Specifically, Pex2 and Pex10 are implicated, along with the Ubc4 E2, in the lysine-targeted polyubiquitination of Pex5, committing it for degradation [51,83]. On the other hand, Pex12 acts with the E2 Pex4 to monoubiquitinate Pex5, priming it for another round of recycling, cargo binding and delivery (figure 2a) [51]. Similar

requirements for Ubc4 and the RING peroxins are observed for Pex18 [53]. By contrast, however, Pex20 seems to require the activity of all three RING peroxins for both its mono- and polyubiquitination and, uniquely, Pex4 participates as an E2 in both processes [54].

### 2.4.3. NY-ESO-1

The NY-ESO-1 protein is a cancer/testis antigen that contains only a single lysine. Ubiquitinated NY-ESO-1 is degraded by the proteasome into antigenic peptides for MHC-I presentation. A recent study demonstrated that removing lysine from NY-ESO-1 did not affect its ability to be effectively recognized and processed by the proteasome. In addition, the authors showed that ubiquitination of lysine-free NY-ESO-1 was sensitive to alkaline treatment, indicating it is occurring via hydroxyester linkages to serine and/or threonine residues [45]. The promiscuous recognition and processing of various types of ubiquitination by the proteasome would help to ensure that antigens of various origins and sequence, whether or not they contain lysine, can still effectively generate peptides for immune display.

### 2.4.4. MARCH1

MARCH1 is an E3 ubiquitin ligase of the membrane-associated RING-CH domain family. Its expression is specific to haematopoietic antigen-presenting cells, such as B cells and resting dendritic cells, where it participates in the regulated ubiquitination and endocytosis of MHC-II and CD86 immune molecules [84]. MARCH1 itself is regulated by strong transcriptional stimulation in response to a resting immune state, which offsets its rapid proteasomal degradation. It was recently demonstrated that MARCH1 with or without its lysine residues is ubiquitinated and degraded by the proteasome [46]. shRNA knockdown of the E2 Ube2D1 strongly stabilized wild-type MARCH1; however, whether this also applies to lysine-free MARCH1 was not addressed. While the responsible E3 ligase(s) has not yet been identified, it appears that MARCH1 itself is not responsible, or is at least not required, as mutation of catalytic residues to disable its E3 activity did not prevent its ubiquitination [46].

## 3. Molecular mechanisms of non-lysine ubiquitination

The examples above demonstrate that non-lysine ubiquitination plays an important cellular function in proteasome-mediated degradation and signalling for proteins involved in many different cellular processes. Still, mechanistic insight into non-lysine ubiquitination has been trailing behind the many observations of its occurrence, and only recently have we begun to better understand the molecular details of how the process takes place. To a large extent, the importance of intrinsic activity of E2 and E3 enzymes towards specific side chains, i.e. lysine versus serine/threonine/cysteine, how such intrinsic activity is accomplished at the molecular level, and how the cooperation between specific E2s and E3s helps to achieve polyubiquitination on non-lysine residues remains unexplored. In this section, we focus on the E2 and E3 enzymes for which at least some of this mechanistic detail is known.

royalsocietypublishing.org/journal/rsob   Open Biol. 9: 190147

royalsocietypublishing.org/journal/rsob   Open Biol. **9**: 190147

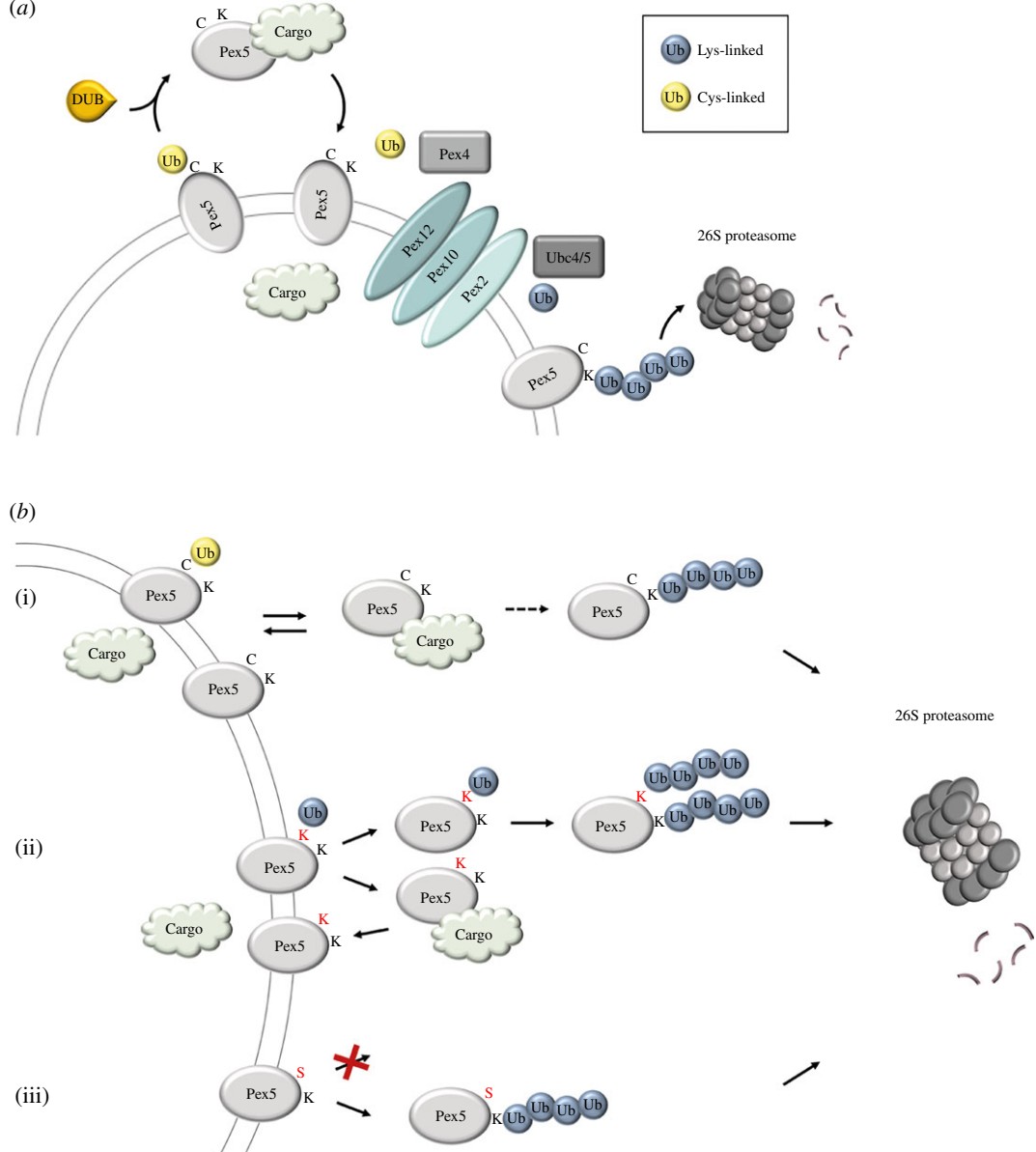

**Figure 2.** Differential ubiquitination pathways influence the fate of Pex5. The figure is based primarily on work performed in *S. cerevisiae* [48,50,51,80]. (*a*) The Pex5 peroxisomal transport receptor is monoubiquitinated on a conserved cysteine (Cys6; indicated by C), enabling its release from the membrane. Upon deubiquitination (DUB; Usp15 in *S. cerevisiae*, Usp9X in mammals; [50]), Pex5 is competent to bind PTS1-bearing cargo and return to the membrane. Pex5 that fails to properly release from or recycle to the membrane is instead polyubiquitinated on nearby lysine residues (Lys18, Lys24; indicated by K) and degraded by the proteasome. Cysteine monoubiquitination is facilitated by the E2 Pex4 (aka Ubc10; UbcH5a/b/c in humans) and the integral membrane RING E3 Pex12, while lysine polyubi-quitination uses Ubc4/5 and RING E3s Pex2 and Pex10. (*b*) The effects of amino acid substitutions on Pex5 regulation. Red labelling of K and S indicates that those are introduced substitutions in place of the naturally occurring conserved C. (i) Wild-type Pex5 gets monoubiquitinated on cysteine and functions normally. A small amount (indicated by the dashed arrow) may partition to polyubiquitination and proteasomal degradation. (ii) The substitution of the conserved cysteine with lysine does not affect the function of Pex5, but does result in lower steady-state levels as a result of increased propensity for polyubiquitination and degradation. (iii) Substitution of the conserved cysteine with serine is unable to support Pex5 function and results in membrane retention, polyubiquitination and degradation. This indicates that not any ubiquitinatable residue can functionally replace Cys6.

## 3.1. The Ube2j2/MARCH6 pair promotes serine/threonine ubiquitination

E3 ligases stimulate the activity of E2 ubiquitin-conjugating enzymes, and dedicated E2/E3 pairs function to facilitate specific cellular reactions [10]. For instance, Ube2j2/mK3, but not Ube2d1/mK3, facilitate non-lysine ubiquitination on the MHC-I HC tail region [28], whereas Ube2j1/Hrd1 preferentially ubiquitinate MHC-I on lysines [28,36]. Simi-larly, the work on Sbh2 [37] and Asi2 [44] demonstrates that a single E3 (Doa10) can promote ubiquitination of both lysine and non-lysine residues depending on the

substrate and the E2 with which it pairs up. These studies suggest that it is at least partly the specific E2/E3 pairing that determines the preferred target site(s) for ubiquitination.

Table 1 shows that among the examples where an E2/E3 pair is known to mediate non-lysine ubiquitination on serine and/or threonine, Ube2j2 and MARCH6 (Ubc6 and Doa10 in yeast) is the prevalent combination observed. In addition to their functional interaction, overexpressed Ube2j2 and MARCH6 have been shown to co-immunoprecipitate [70]. Given the apparent importance of the Ube2j2/MARCH6 pair, we will examine this in more detail below.

In *S. cerevisiae*, recent work has shown that Ubc6 works in tandem with Ubc7 (the yeast orthologue of metazoan Ube2g2) in polyubiquitinating substrates of Doa10 [37]. In cells devoid of Ubc7, a monoubiquitinated ERAD substrate became apparent, and only in the presence of both E2s was strong Lys48-linked polyubiquitination, mediated by Ubc7, observed. The same conclusions were drawn based on *in vitro* experiments, where it was also found that Ubc6 monoubiquitinates on a serine residue. Using a lysine-free mutant of a known ERAD substrate, the tail-anchored Sbh2 protein that is destabilized in the absence of its interactor Ssh1, it was also demonstrated that substrate turnover strongly relied on Ubc6 and that polyubiquitination of this substrate was sensitive to NaOH treatment.

This work illustrates the important common mechanism of ubiquitin priming by one E2 enzyme followed by chain elongation by another E2 that dictates ubiquitin chain topology, as previously observed in various cellular systems (reviewed in [10]). Thus, the Ubc6/Doa10 pair primes the substrate by addition of a single ubiquitin molecule (e.g. on serine), and then Ubc7/Doa10 adds sequential Lys48-linked ubiquitin molecules to target the substrate for degradation. As described above, the Ube2j2/TMEM129 pair has also been suggested to prime the cytosolic tail of MHC-I HC by serine/threonine ubiquitination prior to Lys48-linked chain elongation by Ube2k (and an unknown E3 ligase) [62]. Furthermore, it was concluded that the Ube2j2/mK3 pair also initiates ubiquitination of the MHC-I HC tail primarily on serine/threonine residues [28]. In this case, though, it was found that the same E2/E3 pair is sufficient to also assemble the Lys48-linked polyubiquitin chain.

## 3.2. Unusual molecular features of Ube2j2 likely determine non-lysine ubiquitination

E2s possess an intrinsic reactivity, which is currently not known to be altered by E3s, towards the N-terminal amino group or various amino acid side chains [10]. For instance, E2s that work together with HECT-type E3 ligases must be able to transfer ubiquitin to cysteine residues. For RING-type E3 ligases, substrate ubiquitination is carried out by the E2 enzyme, which must, therefore, have intrinsic reactivity towards its specific target(s). Using free amino acids as nucleophiles, most investigated E2s that work with RING-type E3s can transfer ubiquitin to both lysine and cysteine. However, Ube2w is specific for the N-terminal amino group [85]. It is also clear that although Ube2j2 (together with mK3) shows a preference for serine/threonine, Ube2j2/mK3 also facilitates lysine ubiquitination [28].

Canonical E2s are characterized by a conserved His-Pro-Asn (HPN) motif in the active site, where the Asn side chain plays an important function in catalysis by interacting with the active-site cysteine. Moreover, it has been proposed that a conserved position in the catalytic cleft, referred to as the conserved E2 Ser (S)/Asp (D) (CES/D) site, divides E2 enzymes into two subgroups: the constitutively active E2s characterized by an aspartic acid residue in the catalytic cleft, and the regulated E2s containing a serine residue in the catalytic cleft, where the serine allows for activation of the E2 enzyme by phosphorylation [86]. Generally, the CES/D site is suggested to be important for aligning the substrate lysine towards the E2 catalytic cysteine to control

ubiquitination efficiency [86]. Notably, Ube2j2 is lacking these canonical E2 signatures (HPN motif and CES/D site) and is, moreover, unusually enriched in basic residues in the catalytic cleft as evident from its crystal structure [87]. In addition, Ube2j2 is lacking the α-helix that follows the CES/D site in canonical E2s. This α-helix is replaced by a longer disordered region of unknown function in Ube2j2.

Altogether, it is likely that the unusual features of the Ube2j2 sequence and structure help determine the activity of the enzyme towards serine and threonine. Still, comprehensive biochemical and structural studies are necessary to establish in detail how Ube2j2, and other E2s, manifest their intrinsic activity towards non-lysine residues.

## 3.3. An E3 ligase with specificity for threonine ubiquitination

While classical RING-type E3 ligases depend on an associated ubiquitin-charged E2 to transfer ubiquitin to the substrate, HECT and RBR-type E3 ligases transfer a thioester-linked ubiquitin directly onto the substrate from an internal cysteine residue (§1.3). Surprisingly, the MYCBP2 E3 ligase was recently found to work by a previously unknown RING-Cys-relay mechanism [47]. Here, ubiquitin is transferred from the RING-associated E2 to a cysteine residue placed on a flexible so-called mediator loop in MYCBP2 and then via intramolecular transthiolation to another E3 cysteine that mediates the final delivery to a substrate threonine or serine residue. Using free amino acids or short peptides containing a central threonine, serine or lysine residue, experiments demonstrated very little lysine reactivity and a clear preference for modification of threonine over serine [47]. Unfortunately, cysteine was not tested as an acceptor in these experiments. As mentioned above (§2.2.3), the cellular MYCBP2 substrate, NMNAT2, undergoes base-labile, thiol-resistant *in vitro* ubiquitination by MYCBP2, underscoring the ability of this E3 ligase to ubiquitinate hydroxyl groups [47].

The catalytic mechanism employed by MYCBP2 in threonine ubiquitination was deduced based on the crystal structure of a catalytic fragment in combination with biochemical assays using point mutants of MYCBP2 [47]. At the active site, a histidine residue works as a general base to extract the threonine hydroxyl proton to facilitate a nucleophilic attack on the thioester-linked ubiquitin. Moreover, a cluster of three phenylalanine residues create a hydrophobic pocket to accommodate the threonine $C^\beta$ methyl group, a feature likely explaining the threonine over serine selectivity by MYCBP2. Taken together, this study provides fascinating insight at a detailed molecular level into the first known example of non-lysine ubiquitination performed by an E3 ligase.

## 4. Biological importance of non-lysine ubiquitination

Ubiquitination on lysine residues regulates a plethora of cellular processes. Why, then, has nature devised non-lysine ubiquitination? The overall answer to this question likely relates to the fact that non-lysine ubiquitination extends the cell biological possibilities of the ubiquitination system beyond those afforded by lysine ubiquitination.

royalsocietypublishing.org/journal/rsob    Open Biol. 9: 190147

## 4.1. Non-lysine ubiquitination extends possibilities for substrate degradation

From the studies described in §2 on lysine-free mutants of, for example, Asi2 and Sbh2 [37,44], it is clear that non-lysine ubiquitination can ensure degradation of substrates that normally rely on lysine-directed ubiquitination for degradation. However, based on all available data, acting as a backup system to ensure degradation of proteins that lose their lysines throughout evolution is hardly a physiological purpose of non-lysine ubiquitination. Rather, the availability of ubiquitination sites on serine, threonine and cysteine serves to extend the possibilities for ubiquitination. Thus, as previously suggested [24,25,37], the finding that priming can take place on residues in addition to lysine likely increases the chances of polyubiquitination by simply creating more sites for chain extension and thus secure more efficient substrate degradation. Polyubiquitinated ERAD substrates are recognized and extracted from the ER by the AAA ATPase p97 [88]. This protein plays a central role in ERAD by unfolding ubiquitin moieties as well as the substrate protein itself to prepare it for degradation by the proteasome [89,90]. Many sites of polyubiquitin attachment will thus increase the chance of p97 recognition and thereby ensure more efficient degradation.

Whether non-lysine ubiquitination is especially important when lysine acceptor sites are sparse in substrates is presently unclear. In this context, it has been pointed out that several AB-type toxins show an unusual paucity of lysines in their sequence [91], with the PTS1 subunit of the pertussis toxin protein being an extreme example, showing an arginine-to-lysine ratio of 22 : 0 [92]. The suggestion that this paucity of lysine residues is caused by an evolutionary pressure to avoid ubiquitination, and hence degradation, is supported by experiments that show decreased cellular activity of these toxins when mutated to contain (additional) lysine residues [57,92]. To the best of our knowledge, similar experiments introducing additional serine, threonine or cysteine residues have not been performed. Neither has a systematic analysis of serine/threonine/cysteine content been published for the AB-type toxins.

It is unlikely, though, that creating more sites for ubiquitination is the only function of non-lysine ubiquitination in degradation. Work on the cytosolic tail of TCRα has shown that replacing the two naturally occurring serine residues that ensure rapid degradation with lysine did not prevent efficient degradation—in fact, ubiquitination levels and degradation kinetics of the overexpressed protein increased for the lysine mutant [33]. While speculative at present, this result could indicate that, for endogenous TCRα, serine ubiquitination allows regulation of the degradation process that is not possible when ubiquitination takes place on lysine.

Such regulation could be influenced by the chemical nature of the hydroxyester versus the isopeptide bond. While the former is chemically more labile, it may at the same time be less sensitive to DUB hydrolysis than isopeptide bonds in ubiquitin–lysine conjugates [28]. To our knowledge, no DUB capable of releasing hydroxyester-linked ubiquitin has been identified to date. This potentially suggests that serine/threonine ubiquitination may serve as a label for commitment to degradation. Given the evidence that many viral targets are ubiquitinated on serine or threonine residues, the absence of hydroxyester-specific DUBs would be advantageous for the virus and a hurdle for the host. For further consideration, if no DUB can cleave the proximal ubiquitin moiety attached to serine/threonine, ubiquitin would have to be degraded together with the substrate (figure 3a). Recent work has demonstrated that the proteasome is, indeed, capable of degrading the proximal ubiquitin along with the substrate [93,94]. Alternatively, the serine/threonine monoubiquitination may serve a signalling function (figure 3a).

## 4.2. Non-lysine ubiquitination facilitates differential regulation of substrate fate

The ability of a single substrate to (i) be ubiquitinated on more than one type of amino acid residue and (ii) potentially require different E2/E3 combinations for isopeptide, hydroxyester or thioester ubiquitin conjugation creates many regulatory possibilities. The fact that these three bond types have different inherent stabilities, with isopeptide being the strongest and thioester the most labile, also presents options for regulation based on the dynamics of ubiquitination versus deubiquitination, be it chemical or enzymatic [24]. Examples of how non-canonical ubiquitination could contribute to substrate regulation are described herein.

The cellular level of the neurogenin (Ngn) transcription factors is regulated in opposing fashion by either binding to a partner E protein, in which case Ngn is stable and active, or, alternatively, being rapidly degraded by the proteasome [95]. Ngn levels may also be regulated in part by phosphorylation, as Ngns are partially stabilized by cyclin-dependent kinase inhibitors. Inappropriate stabilization of Ngns correlates with promotion of neurogenesis, highlighting the importance of its careful regulation. As such, it is perhaps not surprising that Ngns are promiscuous ubiquitination substrates that can be modified on the N-terminus as well as on internal lysine, cysteine or serine/threonine residues. All modifications were not, however, equally efficient at supporting degradation. For Ngn3, removing only lysine residues had a dramatic effect on half-life while removing only cysteines resulted in no significant difference [42]. By contrast, the rapid degradation of Ngn2 in mitotic extract was similarly compromised by the absence of either lysines or cysteines [41]. Interestingly, there was no effect of cysteine mutation on Ngn2 stability in interphase extract, suggesting that the ubiquitination of cysteines may be used, perhaps in a cell cycle-dependent manner, as a stability regulation mechanism [41].

Studies of MHC-I HC demonstrated that lysine residues or serine/threonine residues could support its ubiquitination and ERAD. The viral mK3 ligase, however, showed a preference for serine/threonine residues even when lysines were available [27,28] (§2.1.1). On the other hand, the cellular E3 ligase HRD1, while able to ubiquitinate MHC-I HC on both lysine and serine/threonine residues, prefers lysine [36]. This suggests that, under normal circumstances, HRD1 may use lysine to turn over MHC-I HC at a rate appropriate to the needs of the cell, but upon viral infection MHC-I HC degradation is inappropriately upregulated by targeted ubiquitination on serine/threonine residues by mK3. The ability of the viral ligase to act with a cellular E2, Ube2J2 [28], to perform non-canonical ubiquitination thus acts to misregulate MHC-I expression at the cell surface.

royalsocietypublishing.org/journal/rsob    Open Biol. **9**: 190147

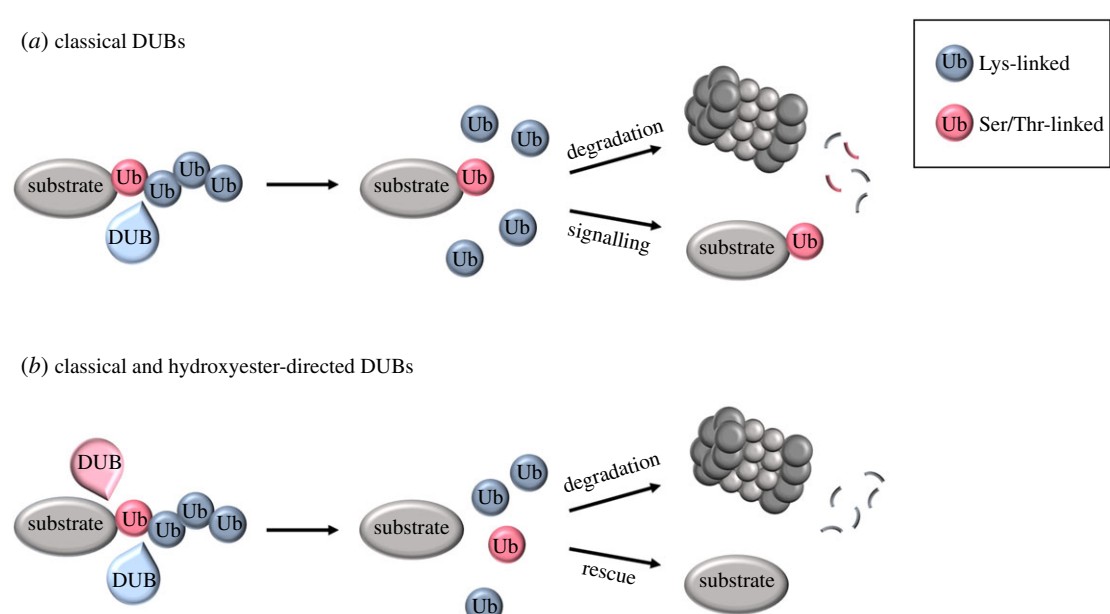

**Figure 3.** Potential DUB regulation of serine/threonine ubiquitination. (*a*) Provided that no hydroxyester-directed DUBs exist, classical DUBs (light blue) will trim the polyubiquitin chain to leave a monoubiquitinated substrate. Degradation will require that ubiquitin is degraded along with the substrate. Potentially, the mono-ubiquitinated substrate could also function in signalling. (*b*) The existence of hydroxyester-directed DUBs (pink) will allow complete ubiquitin removal. This will lead to either proteasomal degradation or substrate rescue.

The studies of peroxisomal systems provide an excellent view into how different combinations of E2s, E3s and target residues for ubiquitination can direct the same protein to, for example, the alternative fates of recycling/reuse or destruction (figure 2). As described in §2.4.2, peroxisome import receptors may be stably polyubiquitinated on lysine and targeted for proteasome-mediated degradation; this is thought to serve a quality control function. Monoubiquitination on a conserved cysteine residue, on the other hand, acts as a transient signal to permit another round of receptor export, cargo binding and return.

The identities of the relevant E2s and E3s for each type of ubiquitination have been worked out and are distinct. Thus, the two opposing fates of peroxisome transport receptors are normally governed by which residues are used for ubiquitination, and by the specific E2(s)/E3(s) involved. An intriguing experiment in which the conserved cysteine was replaced with a lysine demonstrated that the Pex5 receptor was still ubiquitinated and, importantly, still able to function correctly; however, its steady-state levels were dramatically reduced [80]. This is likely due to the increased stability of the isopeptide linkage to lysine, which persists and becomes polyubiquitinated, targeting excessive amounts of Pex5 for proteasomal degradation (figure 2b(ii)). As perturbations in Pex5 levels would affect peroxisome homeostasis, which is linked to numerous human disease states, the importance of correctly regulating Pex5 levels is clear [96]. This highlights the critical nature not only of having an amino acid capable of accepting ubiquitin at a specific position, but also that it must support both proper function and proper regulation of the target protein. In keeping with this, the replacement of the conserved cysteine with serine prevented monoubiquitination, release and recycling of Pex5 and instead resulted in its membrane retention, polyubiquitination and degradation (figure 2b(iii)) [50].

Yet another way in which the steady-state levels of substrates ubiquitinated on varying amino acids could be regulated is by the presence or absence of ubiquitin hydrolases capable of acting upon them. As mentioned above, very little is yet known about the DUBs for non-canonical ubiquitin conjugates. It has been shown, however, that the *S. cerevisiae* ubiquitin-specific protease Usp15, which localizes partially to peroxisomes, can act upon poly- or monoubiquitinated Pex5, implying that it can attack both isopeptide and thioester ubiquitin conjugates [97]. Furthermore, researchers using a biochemical approach identified mammalian Usp9X (homologous to Usp15) as a DUB for cysteine monoubiquitinated Pex5. Again, however, this activity was not specific to thioester linkages as the mutation of the conserved cysteine to lysine still resulted in monoubiquitinated populations of Pex5 that were acted upon by Usp9X [98]. The identification of DUBs capable of acting upon hydroxyester linkages would illuminate additional mechanisms for substrate regulation (figure 3b) and argue against the proposal that serine/threonine ubiquitination serves as a commitment to degradation.

## 5. Concluding remarks

The steady increase in documented cases of non-lysine ubiquitination in recent years testifies to the cell biological importance of the process. Non-lysine ubiquitination takes place in different organisms, at various cellular locations, and regulates a wide range of cellular processes. Moreover, non-lysine ubiquitination should not be seen merely as a 'supplement' to lysine ubiquitination for lysine-free proteins since ample evidence shows that certain E2/E3 pairs preferentially ubiquitinate non-lysine residues.

Despite the growing awareness of the importance of non-lysine ubiquitination many open questions exist. It will thus be interesting to learn whether further examples of non-lysine ubiquitination in cellular signalling, in addition to the regulation of peroxisomal proteins, exist. In some cases of RING-type E3-mediated non-lysine ubiquitination, serine seems to be preferred over threonine. While the intrinsic activity of the involved E2s may govern serine versus

threonine preference, the underlying cause is still unclear. It is also unresolved to which extent the position and local environment of the ubiquitinated residue plays a role in determining ubiquitination efficiency. To date, these questions have only been addressed in a few cases [33,99,100]. The following questions, among others, also remain unanswered. Is there competition for or cross-talk between serine/threonine ubiquitination and phosphorylation? Does cysteine ubiquitination compete with the many possible modifications known to take place on cysteine side chains? Both scenarios present even greater opportunities for regulatory control. Does MYCBP2 constitute a lone example of an E3 ligase with non-lysine substrate specificity or are we in for more surprises? What structural and biochemical features determine intrinsic reactivity of E2 enzymes towards non-lysine residues and how are they activated by their cognate E3s? Do DUBs capable of hydrolysing hydroxyester bonds exist?

Answering certain of these questions will necessitate the development of better experimental tools to determine sites of cellular non-lysine ubiquitination. While alkaline treatment (for serine/threonine ubiquitination) or reduction (cysteine ubiquitination) as well as mutational analysis are informative strategies, high-resolution methods are needed to determine specific sites of ubiquitination. Here, MS would be the method of choice, but unfortunately has been unsuccessful in certain cases [35,46], potentially due to the labile nature of the hydroxyester bond. However, it is encouraging that recent investigations have been more successful in employing MS for the identification of serine ubiquitination [37,39]. To date, global analyses of the ubiquitome have successfully enriched samples for ubiquitinated lysine residues through the use of antibodies specifically raised against a protein antigen containing diglycine-modified lysines (anti-K-ε-GG; [101]). The generation of antibodies that serve this purpose for ubiquitin-modified serines, threonines and cysteines would provide a means to examine how pervasive non-lysine ubiquitination may be.

When surveying the literature on non-lysine ubiquitination it becomes apparent that the vast majority of studies often discovered serine, threonine or cysteine ubiquitination fortuitously during the course of investigating a specific protein. On the contrary, few studies have investigated non-lysine ubiquitination *per se* with the goal of learning more about the molecular mechanism of the process itself. With the information now available, it will next be critical to ask targeted questions that examine the molecular mechanisms of non-lysine ubiquitination in a cellular context. In parallel, it will be essential to undertake biochemical and structural investigations of the central E2 and E3 enzymes implicated in this process, both in isolation and together. Perhaps we can then better understand how they cooperate in performing their functions and gain molecular insight at the atomic level into the mechanisms of non-lysine ubiquitination.

Data accessibility. This article does not contain any additional data.
Authors' contributions. A.J.M., S.H.L. and L.E. all drafted and revised the manuscript. All authors gave final approval for publication.
Competing interests. We declare we have no competing interests.
Funding. L.E.'s work on this project was funded by the Department of Biology, University of Copenhagen. A.J.M.'s work on this project was supported by Bennington College.

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
