## [Reviewer comments · Open Biology]

Review History

RSOB-19-0147.R0 (Original submission)

Review form: Reviewer 1

Recommendation

Accept with minor revision (please list in comments)

Do you have any ethical concerns with this paper?

No

Comments to the Author

In addition to the well-studied lysine ubiquitination that is central to many biological functions, there are multiple examples of ubiquitination on non-lysine residues including cysteines, serines and threonines. This is a timely and very thorough review on the mechanism of non-lysine ubiquitination and its cellular functions. It is also quite assessible. I have just a few minor suggestions and comments that the authors might consider adding.

1. Since there is significant information available on non-lysine Ub in ERAD, it would be nice to include a figure on ERAD and include the role of p97, which I do not believe is mentioned. The

fact that Ub modification of soluble proteins occurs as they emerge in the cytosol and the tails of membrane proteins can be rather short, it might be reasonable to suggest that the variety of amino acids that can be ubiquitinated might enhance the chance that misfolded ER proteins can be modified and thus recognized by p97, which is central to the extraction of almost every ERAD client that has been studied.

2. There is no mention of the toxins that enter the cytoplasm by being recognized as ERAD clients. Many of these proteins are devoid of lysines, which makes them an interesting exception in the context of this review. Perhaps this could be added to the concluding remarks at the end of the review.

3. There are now data to demonstrate that the TCR α chains that are discussed are not integral membrane proteins when they remain unassembled. Instead the charges that are critical to assembly with CD3 subunits prevent their integration resulting in them localizing to the ER lumen where they encounter the ER chaperones that target them for degradation. Maybe this part could be moved to the section on luminal ERAD clients.

Review form: Reviewer 2 (Paul J. Lehner)

Recommendation

Accept with minor revision (please list in comments)

Do you have any ethical concerns with this paper?

No

Comments to the Author

This was a well-written, comprehensive and timely review on the subject of non-lysine mediated ubiquitination.

I have a few, relatively minor comments and questions:

1 Is it correct to assume that the unbiased, systematic approaches to defining the ubiquitome, will have missed the non-lysine targets – and therefore these will not be curated in any ubiquitin proteome database – either way it would be helpful to have this topic discussed?

2 It's remarkable how many of the viral targets appear to use non-lysine hydroxyester bonds. This might be advantageous to the virus if there are indeed no hydroxyester-specific DUBs? This might be worth discussing?

3 in the past decade – in abstract – first demonstration was in fact in 2005 – seems longer than a decade?

4 Line 62 The 19S cap also contains deubiquitinating enzymes (DUBs), of which approximately 100 have been described to date [2] – try to disambiguate this sentence. I understand what they mean, but they don't wish to give the impression that there are ~100 DUBs on the proteasome..

5 Line 67 Moreover, ubiquitin removal by DUBs can provide a last way out for proteins otherwise destined for proteasomal degradation [4, 5] – unclear what is meant here.

6 Line 80 uses the energy supplied by ATP hydrolysis

7 Line 377 BST2 is indeed the correct name, but it might also be useful to inform the reader that it is often known by its other name i.e. tetherin?

8 Should be mentioned that Ube2j2 is itself a membrane embedded E2?

9 Line 623 'To our knowledge no DUB capable of releasing hydroxyester-linked ubiquitin has been identified to date. This potentially suggests that serine/threonine ubiquitination may serve as a label for commitment to degradation'.....Might be worth speculating that this is why it is commonly used by viruses?

10 Line 719 Current evidence suggests that for RING type E3-mediated non-lysine ubiquitination, serine seems to be preferred over threonine. What is the evidence for this?

Decision letter (RSOB-19-0147.R0)

30-Jul-2019

Dear Dr Ellgaard,

We are pleased to inform you that your manuscript RSOB-19-0147 entitled "Cellular functions and molecular mechanisms of non-lysine ubiquitination" has been accepted by the Editor for publication in Open Biology. The reviewer(s) have recommended publication, but also suggest some minor revisions to your manuscript. Therefore, we invite you to respond to the reviewer(s)' comments and revise your manuscript.

Please submit the revised version of your manuscript within 7 days. If you do not think you will be able to meet this date please let us know immediately and we can extend this deadline for you.

1) A text file of the manuscript (doc, txt, rtf or tex), including the references, tables (including captions) and figure captions. Please remove any tracked changes from the text before submission. PDF files are not an accepted format for the "Main Document".

2) A separate electronic file of each figure (tiff, EPS or print-quality PDF preferred). The format should be produced directly from original creation package, or original software format. Please note that PowerPoint files are not accepted.

3) Electronic supplementary material: this should be contained in a separate file from the main text and meet our ESM criteria (see <http://royalsocietypublishing.org/instructions-authors#question5>). All supplementary materials accompanying an accepted article will be treated as in their final form. They will be published alongside the paper on the journal website and posted on the online figshare repository. Files on figshare will be made available approximately one week before the accompanying article so that the supplementary material can be attributed a unique DOI.

Online supplementary material will also carry the title and description provided during submission, so please ensure these are accurate and informative. Note that the Royal Society will not edit or typeset supplementary material and it will be hosted as provided. Please ensure that the supplementary material includes the paper details (authors, title, journal name, article DOI). Your article DOI will be 10.1098/rsob.2016[*last 4 digits of e.g. 10.1098/rsob.20160049*].

4) A media summary: a short non-technical summary (up to 100 words) of the key findings/importance of your manuscript. Please try to write in simple English, avoid jargon, explain the importance of the topic, outline the main implications and describe why this topic is newsworthy.

Images

Data-Sharing

It is a condition of publication that data supporting your paper are made available. Data should be made available either in the electronic supplementary material or through an appropriate repository. Details of how to access data should be included in your paper. Please see <http://royalsocietypublishing.org/site/authors/policy.xhtml#question6> for more details.

Data accessibility section

Sincerely,

The Open Biology Team
<mailto:openbiology@royalsociety.org>

Reviewer(s)' Comments to Author:

Referee: 1

Comments to the Author(s)

In addition to the well-studied lysine ubiquitination that is central to many biological functions, there are multiple examples of ubiquitination on non-lysine residues including cysteines, serines and threonines. This is a timely and very thorough review on the mechanism of non-lysine ubiquitination and its cellular functions. It is also quite assessible. I have just a few minor suggestions and comments that the authors might consider adding.

1. Since there is significant information available on non-lysine Ub in ERAD, it would be nice to include a figure on ERAD and include the role of p97, which I do not believe is mentioned. The fact that Ub modification of soluble proteins occurs as they emerge in the cytosol and the tails of membrane proteins can be rather short, it might be reasonable to suggest that the variety of amino acids that can be ubiquitinated might enhance the chance that misfolded ER proteins can be modified and thus recognized by p97, which is central to the extraction of almost every ERAD client that has been studied.
2. There is no mention of the toxins that enter the cytoplasm by being recognized as ERAD clients. Many of these proteins are devoid of lysines, which makes them an interesting exception in the context of this review. Perhaps this could be added to the concluding remarks at the end of the review.
3. There are now data to demonstrate that the TCRA chains that are discussed are not integral membrane proteins when they remain unassembled. Instead the charges that are critical to assembly with CD3 subunits prevent their integration resulting in them localizing to the ER lumen where they encounter the ER chaperones that target them for degradation. Maybe this part could be moved to the section on luminal ERAD clients.

Referee: 2

Comments to the Author(s)

This was a well-written, comprehensive and timely review on the subject of non-lysine mediated ubiquitination.

I have a few, relatively minor comments and questions:

- 1 Is it correct to assume that the unbiased, systematic approaches to defining the ubiquitome, will have missed the non-lysine targets – and therefore these will not be curated in any ubiquitin proteome database – either way it would be helpful to have this topic discussed?
- 2 It's remarkable how many of the viral targets appear to use non-lysine hydroxyester bonds. This might be advantageous to the virus if there are indeed no hydroxyester-specific DUBs? This might be worth discussing?
- 3 in the past decade – in abstract – first demonstration was in fact in 2005 – seems longer than a decade?
- 4 Line 62 The 19S cap also contains deubiquitinating enzymes (DUBs), of which approximately 100 have been described to date [2] – try to disambiguate this sentence. I

understand what they mean, but they don't wish to give the impression that there are ~100 DUBS on the proteasome..

5 Line 67 Moreover, ubiquitin removal by DUBs can provide a last way out for proteins otherwise destined for proteasomal degradation [4, 5] – unclear what is meant here.

6 Line 80 uses the energy supplied by ATP hydrolysis

7 Line 377 BST2 is indeed the correct name, but it might also be useful to inform the reader that it is often known by its other name i.e. tetherin?

8 Should be mentioned that Ube2j2 is itself a membrane embedded E2?

9 Line 623 'To our knowledge no DUB capable of releasing hydroxyester-linked ubiquitin has been identified to date. This potentially suggests that serine/threonine ubiquitination may serve as a label for commitment to degradation'.....Might be worth speculating that this is why it is commonly used by viruses?

10 Line 719 Current evidence suggests that for RING type E3-mediated non-lysine ubiquitination, serine seems to be preferred over threonine. What is the evidence for this?

Author's Response to Decision Letter for (RSOB-19-0147.R0)

See Appendix A.

Decision letter (RSOB-19-0147.R1)

27-Aug-2019

Dear Dr Ellgaard

We are pleased to inform you that your manuscript entitled "Cellular functions and molecular mechanisms of non-lysine ubiquitination" has been accepted by the Editor for publication in Open Biology.

Sincerely,

The Open Biology Team
 mailto: openbiology@royalsociety.org

Appendix A

Cellular functions and molecular mechanisms of non-lysine ubiquitination

Amie J. McClellan et al.

Point-by-point response to reviewers:

Reviewer #1, overall comment:

In addition to the well-studied lysine ubiquitination that is central to many biological functions, there are multiple examples of ubiquitination on non-lysine residues including cysteines, serines and threonines. This is a timely and very thorough review on the mechanism of non-lysine ubiquitination and its cellular functions. It is also quite assessible. I have just a few minor suggestions and comments that the authors might consider adding.

Reviewer #1, comment 1:

Since there is significant information available on non-lysine Ub in ERAD, it would be nice to include a figure on ERAD and include the role of p97, which I do not believe is mentioned. The fact that Ub modification of soluble proteins occurs as they emerge in the cytosol and the tails of membrane proteins can be rather short, it might be reasonable to suggest that the variety of amino acids that can be ubiquitinated might enhance the chance that misfolded ER proteins can be modified and thus recognized by p97, which is central to the extraction of almost every ERAD client that has been studied.

Answer:

As pointed out, p97 plays a central role in ERAD by extracting substrates from the ER, a function mediated by binding to ubiquitin moieties on polyubiquitinated substrates. In our manuscript, we do mention that the variety of amino acids that can be ubiquitinated may enhance the chance that misfolded ER proteins can be modified (lines 615-618):

“Thus, as previously suggested [24, 25, 60], the finding that priming can take place on residues in addition to lysine likely increases the chances of polyubiquitination by simply creating more sites for chain extension and thus secure more efficient substrate degradation.”

To acknowledge the important role of p97 in ERAD, we have added the following text directly after the text cited above:

“Polyubiquitinated ERAD substrates are recognized and extracted from the ER by the AAA ATPase p97 [87]. This protein plays a central role in ERAD by unfolding ubiquitin moieties as well as the substrate protein itself to prepare it for degradation by the proteasome [88, 89]. Many sites of polyubiquitin attachment will thus increase the chance of p97 recognition and thereby ensure more efficient degradation”.

However, our review focuses on non-lysine ubiquitination and not on the ERAD process as such, and we have not been able find any direct connection between non-lysine ubiquitination and p97 function. Therefore, we have chosen not to include an ERAD figure (including the role of p97) in the revised manuscript.

Reviewer #1, comment 2:

There is no mention of the toxins that enter the cytoplasm by being recognized as ERAD clients. Many of these proteins are devoid of lysines, which makes them an interesting exception in the context of this review. Perhaps this could be added to the concluding remarks at the end of the review.

Answer:

We thank the reviewer for bringing up this important point, which clearly deserves attention.

To introduce the topic, we have added the following sentence (line 192-195):

“Not only viruses exploit the ERAD pathway for their own purposes – a range of AB type toxins, such as cholera, ricin and pertussis, use the ERAD pathway to gain access to the cytosol, where they perform their deleterious functions ([31]; Section 4.1).”

As mentioned by the reviewer, several of the AB type toxins show a paucity of lysines, presumably as a mechanism to avoid degradation. These findings have apparently never been put into context of non-lysine ubiquitination.

In the revised manuscript, we now address this point after the sentence: “Whether non-lysine ubiquitination is especially important when lysine acceptor sites are sparse in substrates is presently unclear”, where we have added the following (lines 625-635):

“In this context, it has been pointed out that several AB type toxins show an unusual paucity of lysines in their sequence [90], with the PTS1 subunit of the pertussis toxin protein being an extreme example, showing an arginine-to-lysine ratio of 22:0 [91]. The suggestion that this paucity of lysine residues is caused by an evolutionary pressure to avoid ubiquitination and hence degradation, is supported by experiments that show decreased cellular activity of these toxins when mutated to contain (additional) lysine residues [31, 91]. To the best of our knowledge, similar experiments where additional serine, threonine or cysteine residues have been introduced, have not been performed. Neither has a systematic analysis of serine/threonine/cysteine content been published for the AB type toxins.”

Reviewer #1, comment 3:

There are now data to demonstrate that the TCR α chains that are discussed are not integral membrane proteins when they remain unassembled. Instead the charges that are critical to assembly with CD3 subunits prevent their integration resulting in them localizing to the ER lumen where they encounter the ER chaperones that target them for degradation. Maybe this part could be moved to the section on luminal ERAD clients.

Answer:

In light of this valuable comment we have now included a reference to the paper by Feige and Hendershot (Mol. Cell (2013), 51(3):297-309, and added the following text at the end of Section 2.1.2 (lines 265-271):

“Notably, unassembled TCR α chains have been demonstrated not to integrate efficiently into the membrane, but instead localize to the ER lumen [46]. In essence, these chains can therefore be considered as soluble ERAD substrates (see below). It is interesting to note that the two very C-terminal (serine) residues become ubiquitinated, indicating that this substrate is retrotranslocated with the C-terminus first and likely becomes ubiquitinated as soon as it enters the cytosol.”

Reviewer #2, overall comment:

This was a well-written, comprehensive and timely review on the subject of non-lysine mediated ubiquitination.

I have a few, relatively minor comments and questions:

Reviewer #2, comment 1:

Is it correct to assume that the unbiased, systematic approaches to defining the ubiquitome, will have missed the non-lysine targets – and therefore these will not be curated in any ubiquitin proteome database – either way it would be helpful to have this topic discussed?

Answer:

This is entirely correct – “unbiased” approaches to defining the ubiquitome are actually not unbiased and, therefore, will miss non-lysine targets. Large-scale approaches, thus far, have been specifically designed to detect ubiquitination only on lysine residues as they utilize an antibody prepared against a protein antigen containing diglycine-modified lysines (anti-K-ε-GG; Xu et al. Global analysis of lysine ubiquitination by ubiquitin remnant immunoaffinity profiling. Nat. Biotechnol., 28 (2010), pp. 868-873). The few examples of ubiquitination on serine verified by mass spec were achieved by isolating only the protein of interest and examining post-trypsin mass spec peaks for those consistent with potential diglycine shifts (GlyGly(K), GlyGly(S), GlyGly(T), and GlyGly(C); Chua, N. K., Hart-Smith, G., Brown, A. J. 2019 Non-canonical ubiquitination of the cholesterol-regulated degron of squalene monooxygenase. J. Biol. Chem., (10.1074/jbc.RA119.007798)). In some cases acetic acid was included to help preserve potential ester bonds (Weber, A., Cohen, I., Popp, O., Dittmar, G., Reiss, Y., Sommer, T., Ravid, T., Jarosch, E. 2016 Sequential Poly-ubiquitylation by Specialized Conjugating Enzymes Expands the Versatility of a Quality Control Ubiquitin Ligase. Mol. Cell. 63, 827-839. (10.1016/j.molcel.2016.07.020)). Altogether, this suggests the need for raising antibodies against post-trypsin protein antigens bearing ubiquitin-modified cysteines, serines, or threonines to permit systematic identification of the non-lysine ubiquitome. We have now added discussion of this (see below) and thank the reviewer for raising this issue so that readers may better understand why we do not currently have a sense of how widespread non-lysine ubiquitination is.

The following text has been added (lines 771-776):

“To date, global analyses of the ubiquitome have successfully enriched samples for ubiquitinated lysine residues through the use of antibodies specifically raised against a protein antigen containing diglycine-modified lysines (anti-K-ε-GG; [100]). The generation of antibodies that serve this purpose for ubiquitin-modified serines, threonines, and cysteines would provide a means to examine how pervasive non-lysine ubiquitination may be.”

Reviewer #2, comment 2:

It's remarkable how many of the viral targets appear to use non-lysine hydroxyester bonds. This might be advantageous to the virus if there are indeed no hydroxyester-specific DUBS? This might be worth discussing?

Answer:

Absolutely. The following text (in bold) has been added to emphasize this point (lines 650-652):

“Given the evidence that many viral targets are ubiquitinated on serine or threonine residues, the absence of hydroxyester-specific DUBs would be advantageous for the virus and a hurdle for the host. For further consideration, if no DUB can cleave the proximal ubiquitin moiety.....”

Reviewer #2, comment 3:

In the past decade – in abstract – first demonstration was in fact in 2005 – seems longer than a decade?

Answer:

Indeed! We have changed this wording to reflect reality so that it now reads: “....., but in the past 15 years...”

Reviewer #2, comment 4:

Line 62: The 19S cap also contains deubiquitinating enzymes (DUBs), of which approximately 100 have been described to date [2] – try to disambiguate this sentence. I understand what they mean, but they don't wish to give the impression that there are ~100 DUBS on the proteasome..

Answer:

This text was definitely misinterpretable. We have attempted to disambiguate and also added a more recent reference on predicted DUB gene numbers. The text (lines 61-64) now reads:

*“The 19S cap also contains two (*S. cerevisiae*) to three (most other eukaryotes) deubiquitinating enzymes (DUBs); over 100 DUB-encoding genes have been predicted in the human genome to date [2, 3]”.*

Reviewer #2, comment 5:

Line 67: Moreover, ubiquitin removal by DUBs can provide a last way out for proteins otherwise destined for proteasomal degradation [4, 5] – unclear what is meant here.

Answer:

Here, we were trying to make the point that deubiquitination at the proteasome could potentially save the protein from being degraded (if the rate of deubiquitination outpaces the rate of substrate binding/unfolding/translocation/degradation). We have clarified the sentence, so that it now reads (lines 68-69; new text in bold):

*“...removal by DUBs **may provide a last-minute opportunity for a protein to escape degradation** [5, 6].”*

Reviewer #2, comment 6:

Line 80: uses the energy supplied by ATP hydrolysis

Answer:

Thank you – “hydrolysis” has been added, line 80

Reviewer #2, comment 7:

Line 377: BST2 is indeed the correct name, but it might also be useful to inform the reader that it is often known by its other name i.e. tetherin?

Answer:

We have now added this information to the text, line 387

Reviewer #2, comment 8:

Should be mentioned that Ube2j2 is itself a membrane embedded E2?

Answer:

We have now added this information to lines 213-214.

Reviewer #2, comment 9:

Line 623: ‘To our knowledge no DUB capable of releasing hydroxyester-linked ubiquitin has been identified to date. This potentially suggests that serine/threonine ubiquitination may serve as a label for commitment to degradation’Might be worth speculating that this is why it is commonly used by viruses?

Answer:

This point was addressed above in response to comment 2.

Reviewer #2, comment 10:

Line 719: Current evidence suggests that for RING type E3-mediated non-lysine ubiquitination, serine seems to be preferred over threonine. What is the evidence for this?

Answer:

Thank you for bringing this up – the evidence is suggestive, but is not comprehensive enough to make this statement at this time. The text has been modified to (line 747-748):

“In some cases of RING type E3-mediated non-lysine ubiquitination, serine seems to be preferred over threonine.”